SYMPOSIUM REVIEW

# Nutrient timing and metabolic regulation

Harry A. Smith and James A. Betts 🆔

*Centre for Nutrition Exercise and Metabolism, Department for Health, University of Bath, Bath, UK*

Edited by: Ian Forsythe & Javier Gonzalez

The peer review history is available in the Supporting Information section of this article (https://doi.org/10.1113/JP280756#support-information-section).

<div style="writing-mode: vertical-rl">The Journal of Physiology</div>

**Abstract** Daily (circadian) rhythms coordinate our physiology and behaviour with regular environmental changes. Molecular clocks in peripheral tissues (e.g. liver, skeletal muscle and adipose) give rise to rhythms in macronutrient metabolism, appetite regulation and the components of energy balance such that our bodies can align the periodic delivery of nutrients with ongoing metabolic requirements. The timing of meals both in absolute terms (i.e. relative

**Harry Smith** (left) is a PhD student in the Centre for Nutrition, Exercise and Metabolism (CNEM) at the University of Bath. **James Betts** (right) is Professor of Metabolic Physiology and co-director of CNEM at Bath. Their research focuses on the role of nutrient timing on metabolic regulation and human health.

This review was an accepted presentation for the American College of Sports Medicine 2020 annual meeting "Novel dietary approaches to appetite regulation, health and performance" which was rescheduled for 2021.

to clock time) and in relative terms (i.e. relative to other daily events) is therefore relevant to metabolism and health. Experimental manipulation of feeding–fasting cycles can advance understanding of the effect of absolute and relative timing of meals on metabolism and health. Such studies have extended the overnight fast by regular breakfast omission and revealed that morning fasting can alter the metabolic response to subsequent meals later in the day, whilst also eliciting compensatory behavioural responses (i.e. reduced physical activity). Similarly, restricting energy intake via alternate-day fasting also has the potential to elicit a compensatory reduction in physical activity, and so can undermine weight-loss efforts (i.e. to preserve body fat stores). Interrupting the usual overnight fast (and therefore also the usual sleep cycle) by nocturnal feeding has also been examined and further research is needed to understand the importance of this period for either nutritional intervention or nutritional withdrawal. In summary, it is important for dietary guidelines for human health to consider nutrient timing (i.e. *when* we eat) alongside the conventional focus on nutrient quantity and nutrient quality (i.e. *how much* we eat and *what* we eat).

(Received 19 September 2021; accepted after revision 9 December 2021; first published online 17 January 2022)

**Corresponding author** J. A. Betts: Centre for Nutrition Exercise and Metabolism, Department for Health, University of Bath, Bath BA2 7AY, UK. Email: J.Betts@bath.ac.uk

**Abstract figure legend** Timing of meal-intake across the day can be considered in both absolute (i.e. clock time) and relative terms (i.e. to other events across the day). In particular meals can be considered relative to predictable cycles of sleep-wake (e.g. nocturnal feeding) and fasting-feeding (e.g. breakfast and intermittent fasting). Likewise, the timing of meal-intake throughout the day can also be considered relative to peaks in the rhythmic control of physiology (e.g. muscle transcript accumulation and/or appetite regulation). Collectively, consideration of these factors provides insight into the complexity of metabolic regulation within the context of nutrient timing.

## Introduction

Life on earth has evolved within the context of a repetitive cycle of *ca* 24 h, whereby environmental variables such as light exposure predictably oscillate during each daily period. As such, natural selection has provided almost all organisms on this planet with endogenous circadian rhythms to help anticipate impending environmental challenges and thus pre-emptively adjust our physiology, metabolism and/or behaviour accordingly (Jagannath *et al.* 2017). The mammalian circadian timing system comprises both a central 'master' clock located in the suprachiasmatic nucleus of the hypothalamus and an integrated network of peripheral clocks located throughout various organs, tissues and cell-types (Albrecht, 2017). Collectively, these molecular clocks facilitate the coordinated disposal, degradation, synthesis and recycling of metabolic substrates in order that our periodic delivery of dietary nutrients (i.e. meal times) can appropriately meet our ongoing physiological requirements (Frayn, 2019). The objective of this review is to briefly summarise the mammalian circadian timing system and the daily rhythmicity of macronutrient metabolism, energy expenditure and appetite regulation, before considering how the alignment of daily feeding patterns with these underlying rhythms can impact human health.

## The mammalian circadian timing system

The suprachiasmatic nucleus can translate repeating environmental stimuli, such as photic input, into the appropriate biological rhythms via a variety of signalling pathways, such as autonomic stimulation, endocrine action and body temperature modification (Lewy *et al.* 1999; Brown *et al.* 2002; Berson, 2003; Buhr *et al.* 2010; Slominski *et al.* 2012). Translation of murine work to humans highlights that molecular regulation of circadian rhythms at a cellular level involves the expression of clock genes, which can maintain approximate 24 h rhythmicity via interlocking transcriptional–translational feedback loops with both positive and negative limbs (Mazzoccoli *et al.* 2012; McGinnis & Young, 2016). The positive limb is characterised by the proteins circadian locomotor output cycles kaput (CLOCK), its paralogue neuronal PAS domain protein 2 (NPAS2), and brain and muscle ARNT-like 1 (BMAL1), which are typically found in the nucleus (Kwon *et al.* 2006). Whilst this positive part of the loop targets clock-controlled genes, it also activates rhythmic transcription within the negative limb, including the *Period* (*PER*) and *Cryptochrome* (*CRY*) genes (Mohawk *et al.* 2012); this serves to inhibit the activity of CLOCK:BMAL1 prior to degradation, thereby ending repression of the positive aspect and initiating a new cycle of transcription (Table 1) (Sahar & Sassone-Corsi, 2012;

**Table 1. Name, definition and basic function of the 'core' circadian clock machinery involved in the transcription–translation feedback loop**

| Name | Definition | Function | Reference |
|---|---|---|---|
| Ebox | Enhancer box | Promoter region that regulates cellular transcriptional activity | Hao *et al*. (1997) |
| RORE | Retinoic acid-related orphan receptor response element. | Promoter region that regulates cellular transcriptional activity | Cook *et al*. (2015) |
| CLOCK | Circadian locomotor output cycles kaput | Forms heterodimer with BMAL1 which binds to and activates the Ebox thereby stimulating transcription and translation of Per and Cry | Buhr & Takahashi (2013) |
| NPAS2 | Neuronal PAS domain protein 2 | Paralogue of CLOCK. Forms heterodimer with BMAL1 which binds to and activates the Ebox thereby activating transcription and translation of Per and Cry | Buhr & Takahashi (2013) |
| BMAL1 (Arntl) | Brain and muscle ARNT-like 1 | Forms heterodimer with CLOCK which binds to and activates the Ebox thereby activating transcription and translation of Per and Cry | Buhr & Takahashi (2013) |
| Cry1,2,3 | Cryptochrome 1, 2, 3 | Form a complex with Period proteins. Inactivates Ebox thereby inhibiting transcription and translation of CLOCK and BMAL1 | Ko & Takahashi (2006) |
| Per1, 2, 3 | Period 1, 2, 3 | Form a complex with cryptochrome proteins. Inactivates Ebox thereby inhibiting transcription and translation of CLOCK and BMAL1 | Ko & Takahashi (2006) |
| NR1D1/2 (REV-ERB$\alpha$/$\beta$) | Nuclear receptor subfamily 1 group D member 1/2 | Repression of *BMAL1* gene expression through binding with RORE sites | Guillaumond *et al*. (2005) |
| ROR-$\alpha$/$\beta$/$\gamma$ | Retinoic acid-related orphan receptors | Transcriptional activator for BMAL1 through binding with RORE sites | Guillaumond *et al*. (2005) |

Buhr & Takahashi, 2013; St John *et al*. 2014). The broad importance of proper circadian alignment is clearly apparent in the expression of this core clock machinery throughout mammalian biology, with 3–16% of all mRNA exhibiting rhythmic daily expression (Mohawk *et al*. 2012; Albrecht, 2017; Dierickx *et al*. 2018).

Circadian rhythmicity is particularly evident in signalling pathways within peripheral tissues that are vital for effective metabolic regulation (e.g. liver, muscle, and adipose tissue) (Fig. 1). Specifically, approximately 6–10% of genes in murine hepatocytes display robust circadian rhythms in a tissue-specific manner, with gene clusters targeting carbohydrate and lipid metabolism (Akhtar *et al*. 2002; Robles *et al*. 2014). Likewise, genome-wide transcriptome analysis of skeletal muscle samples from humans reveals high amplitude oscillations for the core clock genes *ARNTL* (*BMAL1*), *NPAS2*, *CLOCK*,

*PER2*, *PER3*, *CRY2*, *NR1D1* (*REV-ERBα*) and *ROR-α* (Perrin *et al*. 2018). Notably, these peaks in transcript accumulation clustered at 16.00 h (for genes implicated in muscle force production and mitochondrial activity) and at 04.00 h (for genes implicated in immune function and inflammation), with rhythmicity also present for genes linked to glucose, lipid and protein homeostasis (Perrin *et al*. 2018). Lastly, approximately 10–20% of the white adipose tissue transcriptome displays 24 h variation, with meaningful temporal oscillations present in both core clock (*PER1*, *PER2*, *PER3*, *CRY2*, *BMAL1* and *DBP*) and metabolic (*REVERBα*, *RIP140* and *PGC1α*) genes under diurnal and constant conditions (Ptitsyn *et al*. 2006; Zvonic *et al*. 2006; Otway *et al*. 2011; Christou *et al*. 2019). Within adipocytes, these core clock genes play an important role in regulating lipolysis, adipogenesis and adipocyte hypertrophy, and so are central to proper

understanding of nutrient balances and obesity (Grimaldi *et al.* 2010; Shimba *et al.* 2011; Guo *et al.* 2012; Paschos *et al.* 2012).

## Rhythms in macronutrient metabolism

With regard to carbohydrate metabolism, whilst basal blood glucose can be relatively elevated upon waking (i.e. the dawn phenomenon), post-prandial glucose tolerance is generally lower in the evening than in the morning (Van Cauter *et al.* 1989, 1992, 1997; Simon *et al.* 1994; Qian & Scheer, 2016). The former is subject to endocrine regulation and driven by hepatic glycogenolysis and gluconeogenesis (Radziuk & Pye, 2006), whereas the latter is primarily regulated by the positive and negative limbs of the transcriptional feedback loop that drives diurnal rhythms in $\beta$-cell responsiveness, insulin secretion/clearance and insulin sensitivity (Baker & Jarrett, 1972; Aparicio *et al.* 1974; Boden *et al.* 1996; Asher *et al.* 2008; Lamia *et al.* 2009; Saad *et al.* 2012; Morris *et al.* 2015*b*; Perrin *et al.* 2018).

By contrast, lipid metabolism favours progressively elevated circulating non-esterified fatty acids, triglyceride and cholesterol later in the day and overnight (Zimmet *et al.* 1974; Morgan *et al.* 1999; Pan & Hussain, 2007; Ang *et al.* 2012; Dallmann *et al.* 2012; Yoshino *et al.* 2014), which is a reflection of diurnal rhythms in lipid storage and mobilisation as opposed to recent

food intake (Yoshino *et al.* 2014; Held *et al.* 2020). Specifically, a combination of animal and human studies suggests that a net shift in fatty acid metabolism from oxidation towards lipogenesis occurs throughout the day, with circadian regulation of intestinal triglyceride absorption, acylcarnitines, mitochondrial oxidative capacity, very-low-density lipoprotein secretion and insulin secretion all contributing to this daily variance (Marrino *et al.* 1987; Lee *et al.* 1992; Pan & Hussain, 2007; Ang *et al.* 2012; Pan *et al.* 2013; Yoshino *et al.* 2014; van Moorsel *et al.* 2016; Sprenger *et al.* 2021).

Finally, in relation to protein metabolism, the majority of amino acids (including all essential, some non-essential and some conditionally essential) display circadian rhythmicity, with peak values occurring between 12.00 and 20.00 h and with lowest values at 04.00–08.00 h (Feigin *et al.* 1967; Wurtman *et al.* 1967; Feigin *et al.* 1968; Grant *et al.* 2019). Variation in the generation and release of amino acids from assorted tissues may underpin this rhythm, including rhythmicity in protein digestion, and absorption (Barattini *et al.* 1993; Fiorucci *et al.* 1995; Qandeel et al. 2009*a*,*b*). The net effect of this variance in amino acid availability on tissue turnover is that protein synthesis is higher during the day and protein oxidation higher at night, with no clear temporal variance in the rate of protein breakdown (Garlick *et al.* 1980; Adam & Oswald, 1981; Kelu *et al.* 2020). This apparent day–night rhythm of muscle protein synthesis is not modulated by the relative absence of dietary protein at night, nor

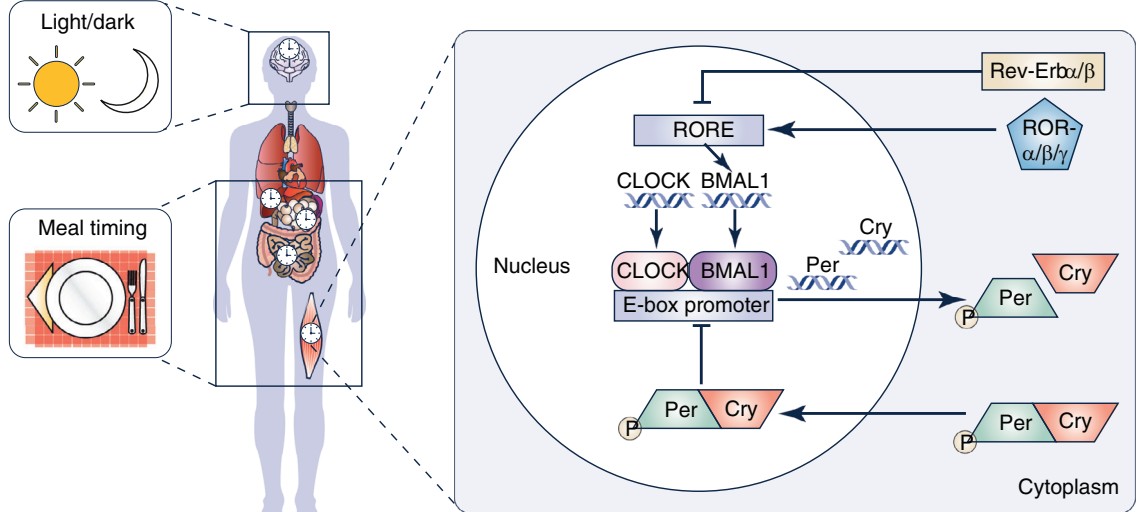

The Journal of
**Physiology**

**Figure 1. The central clock is located in the brain in the suprachiasmatic nucleus (SCN) and is robustly driven by regular cycles of light and dark**
Core clock machinery is also present in numerous metabolically important peripheral tissues such as the liver, skeletal muscle, adipose tissue and gut. Given the role of these tissues in processing ingested nutrients, it is perhaps unsurprising that the effects of meal timing on metabolism are mediated by these peripheral clocks.

the overnight endocrine response (Beelen *et al.* 2008; Betts *et al.* 2011) but is consistent with the rhythmic regulation of MyoD (a myogenic transcription factor) by the CLOCK:BMAL1 complex (Andrews *et al.* 2010; Perrin *et al.* 2018).

## Rhythms in energy expenditure

In stark contrast to the periodic arrival of dietary nutrients from regular daily meals, our ongoing metabolic requirements present a relentless need for continuous energy expenditure. Nonetheless, although unceasing, the rate of thermogenesis also exhibits variability over time and is integral to circadian regulation. For example, elevated body temperature is generally observed during daylight/waking hours, with lower temperature coincident with the dark/sleeping phase amongst most humans, which contributes to synchronising central and peripheral clock machinery (Edwards *et al.* 2002; Buhr *et al.* 2010). Indeed, constant routine protocols (removal of environmental/behavioural stimuli through prolonged wakefulness and even distribution of energy intake) reveal that heat production, oxygen uptake ($\dot{V}_{O_2}$), and carbon dioxide production ($\dot{V}_{CO_2}$) are all highest during the biological morning (Krauchi & Wirz-Justice, 1994; Spengler *et al.* 2000), whereas a recent forced desynchrony protocol (non-standard daily behavioural patterns under dim light conditions) demonstrated that resting metabolic rate is lowest during the late biological night and highest $\sim$12 h later in the biological afternoon/evening (Zitting *et al.* 2018). Interestingly resting energy expenditure also changes overnight with differing stages of sleep (as assessed by sleep encephalography). Generally energy expenditure tends to be highest during lighter/earlier phases, and lowest during the deepest/later stages of sleep (Brebbia & Altshuler, 1965; Fontvieille *et al.* 1994), but some studies have failed to replicate any differences between stages of sleep (Webb & Hiestand, 1975; Haskell *et al.* 1981; White *et al.* 1985; Palca *et al.* 1986; Jung *et al.* 2011). Beyond basal metabolic requirements (i.e. under fasted and resting conditions), an endogenously driven daily rhythm has been reported in diet-induced thermogenesis (i.e. the thermic effect of feeding), with $\sim$20–44% higher values in the morning relative to the evening (Romon *et al.* 1993; Bo *et al.* 2015; Morris *et al.* 2015*a*). However, recent evidence indicates that this is apparent rhythmicity in diet-induced thermogenesis can be accounted for by the underlying circadian variation in resting metabolic rate (Ruddick-Collins *et al.* 2021). Finally, although highly individual, a range of contrasting diurnal patterns of physical activity thermogenesis have been identified, with more intense physical activity often favoured earlier in the day (Maddison *et al.* 2009; Sartini *et al.* 2015; Jansen *et al.* 2018).

## Rhythms in appetite regulation

Evidence for circadian rhythms in appetite and appetite regulatory peptides has been generated using experimental protocols involving both the constant routine and forced desynchrony protocols introduced above. These studies have revealed that hunger is typically lowest in the morning ($\sim$08.00 h) and peaks in the evening ($\sim$20.00 h), when satiety also tends to be lowest (Scheer *et al.* 2013; Sargent *et al.* 2016; Rynders *et al.* 2020; Templeman *et al.* 2021*b*). This robust rhythmicity in appetite ratings occurs independent of time since waking, inter-meal intervals and the energy content of meals (Scheer *et al.* 2013), but is nonetheless entirely consistent with the typical feeding pattern in westernised societies, whereby energy intake tends to be lowest in the morning and highest in the evening (NHANES, 2016).

Our recent work employed a semi-constant routine (i.e. continuous feeding throughout waking hours) to examine the 24 h profile of appetite regulatory hormones (Templeman *et al.* 2021*b*). In that study we reported diurnal rhythms in leptin (peak 00.32 h) and unacylated ghrelin (peak 08.26 h) (Fig. 2). Notably, despite nominally being classified as a hunger hormone, the observed rhythm of ghrelin was approximately antiphasic with that of subjective hunger and ratings of prospective food consumption, which peaked as expected in the evening (i.e. *ca* 20.00–21.00 h) – this phase separation between peaks in appetite ratings and appetite hormones was also evident in another recent study (Rynders *et al.* 2020). In addition to leptin and ghrelin, such daily rhythmicity has also been identified for other appetite regulatory peptides, such as: glucagon-like peptide-1 (peak $\sim$10.00 h, nadir $\sim$17.00 h), peptide YY (peak at $\sim$14.00 h, nadir

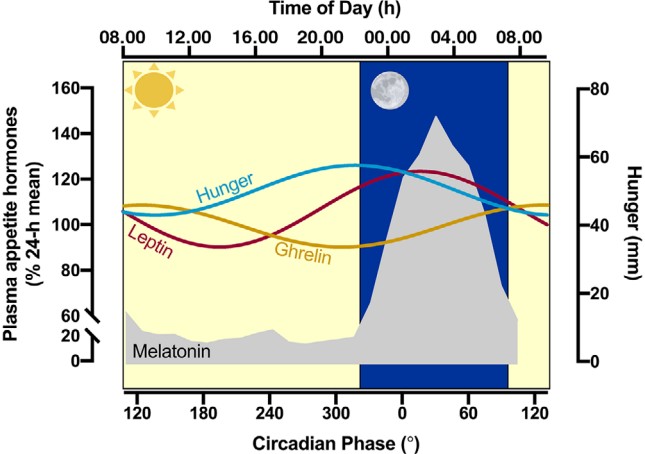

**Figure 2. Diurnal profiles of hunger**
Diurnal rhythms in unacylated ghrelin, leptin and subjective hunger under conditions of semi-constant routine (i.e. hourly feeding during waking hours only) relative to melatonin profile (grey) and light/dark (yellow/blue respectively)

∼04.00 h) and pancreatic polypeptide (peak ∼15.00 h, nadir ∼09.00 h) (Johns *et al.* 2006; Hill *et al.* 2011; Galindo Munoz *et al.* 2015; Rynders *et al.* 2020).

## Nutrient timing

Endogenously controlled rhythms are entrained to environmental time cues known as zeitgebers or 'time givers' (Aschoff, 1954; Aschoff & Pohl, 1978). These include naturally repeating cycles of light–dark, waking–sleeping and activity–rest but also our transitions between the fed–fasted state. As such, the scheduling/alignment of eating occasions (i.e. chrono-nutrition; Flanagan *et al.* 2021), and thus the availability of exogenous nutrients, relative to other regular daily events can serve as a powerful signal to help entrain the endogenous rhythms described in the previous sections (la Fleur *et al.* 2001; Zambon *et al.* 2003; Duffy & Czeisler, 2009; Figueiro *et al.* 2012; Leproult *et al.* 2014; Cheung *et al.* 2016; Tanaka *et al.* 2020). Therefore, in addition to the conventional focus of dietary guidelines for human health regarding nutrient quantity and nutrient quality (i.e. *how much* we eat and *what* we eat), it is also important to consider nutrient timing (i.e. *when* we eat).

Nutrient timing can be understood in terms of both absolute timing (i.e. objective time-of-day, clock time) and relative timing (i.e. with respect to when other relevant events occur and/or usually occur, e.g. wake/sleep, exercise, other meals). The physiological responses to identical meals consumed at different times of day can vary dramatically. For example, as noted earlier, carbohydrate, lipid and protein metabolism all exhibit marked morning–evening differences (Van Cauter *et al.* 1992; Yoshino *et al.* 2014; Morris *et al.* 2015*a*; Leung *et al.* 2019), yet the complete absence of daily food intake for 24 h (i.e. fasting) can eradicate the circadian rhythm in hepatic gene expression that would otherwise occur with a regular meal pattern (Vollmers *et al.* 2009). Even just a short delay in habitual meal timing can alter *Per2* phase in adipose tissue, with corresponding phase shifts in systemic metabolites and hormones but without altering the temporal pattern of melatonin or cortisol (robust markers of the central clock) – all consistent with the idea that peripheral rhythms are closely matched to the absolute time of feeding each day (Schoeller *et al.* 1997; Wehrens *et al.* 2017; Gu *et al.* 2020). Indeed, feeding responsive hormones such as insulin, glucagon and insulin-like growth factor 1 appear to be especially potent modulators of clock gene and/or protein expression in multiple tissues – at least in murine models, but emerging evidence is now beginning to demonstrate this in humans (Tahara *et al.* 2010; Mukherji *et al.* 2015; Sun *et al.* 2015; Ikeda *et al.* 2018; Crosby *et al.* 2019; Tuvia *et al.* 2021).

**Extended overnight fasting.** In terms of relative nutrient timing, the 'other relevant events' that can both influence and be influenced by the response to feeding may include light exposure, sleeping, exercise and, critically, other eating occasions. Breakfast is an eating occasion with particular potential to serve as a zeitgeber and to modify subsequent responses, since this first meal of the day generally marks the end of the overnight period of darkness, sleeping, resting and fasting, whilst also preceding all other daily events. The capacity of breakfast to exert a marked influence on metabolic control later in the day is perhaps best illustrated by the 'second-meal effect', which describes how the glycaemic and insulinaemic responses to repeated carbohydrate ingestion are attenuated relative to an initial meal hours earlier (Hamman & Hirschman, 1919). This phenomenon was first observed using sequential oral glucose tolerance tests but has since been replicated with intravenous infusions (Szabo *et al.* 1969) and mixed macronutrient breakfasts relative to extended morning fasting (Gonzalez, 2014; Chowdhury *et al.* 2015, 2016*b*; Jakubowicz *et al.* 2017). Interestingly the availability of systemic glucose across the morning has been suggested as a possible determinant of physical activity levels in breakfast 'consumers' relative to 'skippers' (Betts *et al.* 2014; Chowdhury *et al.* 2016*a*). Whilst the precise mechanisms underpinning the second-meal effect remain the subject of current investigations (Lee *et al.* 2011; Edinburgh *et al.* 2017; Edinburgh *et al.* 2018), the study by Jakubowicz *et al.* (2017) supports that maintenance of rhythmic clock gene expression plays a role, since the expected pattern of core clock gene expression in leukocytes is disrupted when habitual breakfast consumers omit their usual morning meal.

Further to the acute metabolic effects of breakfast on the responses to subsequent meals later within the same day, recent research has also explored the longer-term effects (i.e. 6 weeks) of regular daily breakfast consumption *versus* extended morning fasting on free-living behavioural responses and any accumulated adaptation in metabolic control. In brief, complete omission of breakfast (i.e. zero energy intake until midday) every day for 6 weeks resulted in significantly lower physical activity thermogenesis than when a regular morning feeding was prescribed – a finding that has been replicated amongst both lean adults and those with obesity (Betts *et al.* 2014; Chowdhury *et al.* 2016*a*). However, other than some evidence in these studies of more stable glycaemia and altered adipose tissue gene expression in lean individuals and improved insulin sensitivity in obese individuals (Gonzalez *et al.* 2018), there were no other effects of regular breakfast on markers of cardiometabolic health nor any metabolic adaptation (Chowdhury *et al.* 2018, 2019). (For a more detailed overview of this series of studies, see Betts *et al.* 2016.)

**Intermittent fasting.** Whilst skipping breakfast is often considered an unhealthy dietary approach (notwithstanding the lack of empirical support for that view), regularly omitting the same meal and/or restricting energy intake to the same set times each day (i.e. time restricted eating) may at least be conducive to the entrainment of endogenous rhythms to that consistently repeating feeding pattern. By contrast, numerous other contemporary approaches to intermittent fasting, often employed as a means to control body weight through weight loss or maintenance, can involve irregular or chaotic patterns of feeding and fasting within each 24 h period (Templeman *et al.* 2020), so are not easily anticipated by the circadian timing system and thus complicate effective metabolic regulation. Popular forms of intermittent fasting within this category include the 5:2 diet (fasting on two non-consecutive days each week) and alternate day fasting (i.e. never feeding on consecutive days). Part of the challenge in understanding the potential effects of any diet based upon intermittent fasting is that the extended periods of complete energy restriction typically culminate in a net energy deficit and therefore weight loss. It therefore becomes difficult to determine whether any observed effects on cardiometabolic health, appetite regulation or other relevant outcomes are attributable to fasting *per se* or simply to the consequences of negative energy balance and reduced adipose tissue mass.

We recently conducted a randomised controlled trial in lean participants expressly to isolate the independent effects of intermittent fasting and net energy restriction (Templeman *et al.* 2021*a*). This was achieved by having some participants impart a prescribed degree of energy restriction but without fasting (i.e. consuming 75% of usual energy intake at each regular meal), whilst others fasted completely every other 24 h but, critically, were re-fed on the alternate days either to match the first group for net energy restriction (i.e. 50% more food than usual on fed days) or to replace the energy 'missed' through fasting altogether (i.e. 100% more food than usual on fed days). Prescribing additional food to minimise or even completely prevent weight loss is understandably not intended to reflect a diet that might be advocated in the real world, but this unusual approach does provide the required experimental design needed to understand the separate and combined effects of fasting and energy (im)balance.

Through the above approach it was possible to determine that standard daily dieting (i.e. without fasting) elicited almost 2 kg of weight loss over 3 weeks and, moreover, that almost all of that change in total body mass was attributable to reductions in body fat content. By contrast, imposing the same prescribed degree of energy restriction via alternate-day fasting resulted in a similar (albeit slightly lower) rate of overall weight loss but this

was accounted for in equal measure by reductions in both fat mass and fat-free mass. Part of the explanation for this apparent difference in energy balance despite ostensibly similar reductions in energy intake is that energy expenditure is not constant but rather has the capacity to compensate for extended periods of fasting to preserve endogenous energy reserves. Specifically, consistent with the adaptive behavioural responses to breakfast omission described earlier, achieving an energy deficit via intermittent fasting can spontaneously inhibit physical activity energy expenditure (i.e. skeletal muscle force production; Westerterp, 2013; Ruddick-Collins *et al.* 2020) to below habitual levels, whereas there was no such change in physical activity levels when the same degree of energy restriction was achieved without fasting (it remains to be seen whether similar behavioural responses occur in obese individuals). Nonetheless, unlike the previously described effect of breakfast omission, there was no difference between any of the interventions in relation to systemic indices of cardiometabolic health, gut hormones, or the expression of key genes in subcutaneous adipose tissue. Overall, the data reported in Templeman *et al.* (2021*a*) further illustrate the complexity of metabolic regulation within the context of nutrient timing since the potential physiological consequences of intermittent fasting may depend upon the interaction between circadian rhythms and related compensatory responses to a modified feeding–fasting pattern.

**Nocturnal interventions.** Excepting the above rather extreme forms of prolonged fasting, most individuals remain in a permanently post-prandial (fed) state for the entirety of daylight/waking hours and so the overnight/sleep phase typically coincides with the longest period of fasting in any given 24 h cycle (Ruge *et al.* 2009). According to the circadian timing system described earlier, this may reflect an entirely natural and properly synchronised alignment between the fed–fasted cycle and all other daily light–dark, wake–sleep and activity–rest cycles. However, it might also be reasoned that this extended period of nutritional withdrawal presents a possible opportunity for dietary intervention. For example, the 'dawn phenomenon' noted earlier highlights how blood glucose may be elevated upon waking, whereas the 'second-meal effect' highlights how prior feeding can be employed to prime the system in preparation for subsequent meals; this begs questions such as whether a nocturnal pre-load can be used to improve glycaemic control in response to breakfast. An initial investigation into such possibilities examined whether waking briefly at 04.00 h to consume a bolus of whey protein might improve metabolic control at breakfast; paradoxically, that nocturnal feeding intervention actually

resulted in impaired glucose tolerance at breakfast, along with elevated lipid oxidation but no effect on appetite (Smith *et al.* 2021). This surprising finding may be partly attributable to the relatively large dose of protein, which was delivered at a time when an abundance of exogenous amino acids is neither required nor expected by the circadian timing system. Consequently, whilst nocturnal feeding presents a possible opportunity for nutritional intervention, it also is a useful paradigm through which we can further understand the relationship between misaligned eating and the increased risk of cardio-metabolic disease.

In addition to balancing the potential benefits and apparent risks of applying nutritional interventions at night, it is also important to consider the indirect impact of interfering with habitual sleep patterns. Indeed, sleep appears to be inherently linked to metabolic regulation, obesity and associated comorbidities, with chronic sleep disorders exerting a potent negative effect on glycaemic control (Briancon-Marjollet *et al.* 2015). For example, to begin with the more extreme model of total sleep deprivation (i.e. remaining awake for one or more nights), fasted glucose concentrations are progressively elevated after 24–120 h of sleeplessness (Kuhn *et al.* 1969; Vondra *et al.* 1981; Wehrens *et al.* 2010; Benedict *et al.* 2011). Post-prandial metabolic control is even more profoundly affected by such models of total sleep restriction, with elevated glycaemic and insulinaemic responses and reduced insulin sensitivity clearly evident after a single night of complete nocturnal wakefulness (Kuhn *et al.* 1969; VanHelder *et al.* 1993; Wehrens *et al.* 2010; Benedict *et al.* 2011). Disrupted sleep may perturb next day metabolism through a multitude of proposed mechanisms; these include, but are not limited to alterations in brain glucose utilization and changes in hormonal secretion profile (Scheen *et al.* 1996), sympathetic nervous stimulation (Spiegel *et al.* 2004), and/or inflammation (Meier-Ewert *et al.* 2004; Vgontzas *et al.* 2004).

Partial sleep deprivation (i.e. a shorter total sleep duration than usual) is a more common occurrence in the real world and can also perturb glycaemic control the following morning, with evidence of impaired glucose clearance and whole-body insulin sensitivity after even a single night of limited sleep (Donga *et al.* 2010; Gonnissen *et al.* 2013; Wang *et al.* 2016; Sweeney *et al.* 2017). Sleep duration can be limited by simply going to bed later and/or getting up earlier or via sleep fragmentation. The latter refers to when sleep is intermittently disrupted by brief waking periods and has the potential to interrupt progression through the various stages of the sleep cycle even if total sleep duration is not substantially curtailed (Tasali *et al.* 2008). We tested the effect of fragmented sleep in our recent work but found post-prandial glucose and insulin responses upon waking to be unaffected by having woken hourly throughout the prior 8-h sleep opportunity (Smith *et al.* 2020). Interestingly, based on the reasoning that a strong coffee is a common remedy following a night of broken sleep, we also investigated the effects of caffeine within the context of the above experimental model. Consistent with the established effects of caffeine on insulin sensitivity independent of sleep deprivation (Robertson *et al.* 2015; Robertson *et al.* 2018), consuming a cup of coffee following a night of sleep fragmentation resulted in a ~50% higher glycaemic response and ~15% higher insulinaemic response at breakfast than either a night of uninterrupted sleep or a matched sleep fragmentation protocol without caffeine prior to breakfast (Smith *et al.* 2020). Further work is therefore needed to better understand whether the potential opportunity for nutritional intervention at night can be harnessed with minimal disruption of sleep patterns, circadian rhythms and next-day metabolic responses.

## Conclusion

Molecular clocks allow for temporal coordination between environmental, metabolic and behavioural cues. Meal patterns are a key element of this system and so considerations regarding nutrient timing should be incorporated into dietary guidelines alongside the conventional focus on nutrient quantity and nutrient quality. Research over the past decade has explored various aspects of nutrient timing and identified several promising approaches to human health improvement involving chrono-nutrition. Further novel insight will be possible through examining the physiological responses of human participants over complete 24-h monitoring cycles, including sequential meal tests, nocturnal feeding and with assessments under free-living conditions.

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

## Additional information

### Competing interests

J.A.B. is an investigator on research grants funded by BBSRC, MRC, British Heart Foundation, Rare Disease Foundation, EU Hydration Institute, GlaxoSmithKline, Nestlé, Lucozade Ribena Suntory, ARLA foods and Kennis Centrum Suiker; has completed paid consultancy for PepsiCo, Kellogg's and SVGC; receives an annual stipend as Editor-in Chief of *International Journal of Sport Nutrition & Exercise Metabolism*; and receives an annual honorarium as a member of the academic advisory board for the International Olympic Committee Diploma in Sports Nutrition.

### Author contributions

Both authors contributed to the writing of this review. Both authors have read and approved the final version of this manuscript and agree to be accountable for all aspects of the work in ensuring that questions related to the accuracy or integrity of any part of the work are appropriately investigated and resolved. All persons designated as authors qualify for authorship, and all those who qualify for authorship are listed.

### Funding

No funding was awarded for the production of this review article.

### Keywords

circadian, meal timing, metabolism, rhythms

### Supporting information

Additional supporting information can be found online in the Supporting Information section at the end of the HTML view of the article. Supporting information files available:

**Peer Review History**

