## [Peer Review History · The Journal of Physiology]

Nutrient Timing and Metabolic Regulation Symposium Review from "Novel dietary approaches to appetite regulation, health and performance (2021)"

Harry A. Smith and James Betts

DOI: 10.1113/JP280756

Corresponding author(s): James Betts (j.betts@bath.ac.uk)

The following individual(s) involved in review of this submission have agreed to reveal their identity: Alexandra Johnstone (Referee #1)

Review Timeline:

Submission Date:	19-Sep-2021
Editorial Decision:	01-Nov-2021
Revision Received:	03-Dec-2021
Accepted:	09-Dec-2021

Senior Editor: Ian Forsythe

Reviewing Editor: Javier Gonzalez

Transaction Report:

Dear Dr Betts,

Re: JP-SR-2021-280756 "Nutrient Timing and Metabolic Regulation Symposium Review from "Novel dietary approaches to appetite regulation, health and performance (2021)"" by Harry A. Smith and James Betts

Thank you for submitting your invited Symposium Review to The Journal of Physiology. It has been assessed by a Reviewing Editor and by 2 expert referees and I am pleased to tell you that it is considered to be acceptable for publication following satisfactory revision.

The reports are copied at the end of this email. Please address all of the points and incorporate all requested revisions, or explain in your Response to Referees why a change has not been made.

NEW POLICY: In order to improve the transparency of its peer review process The Journal of Physiology publishes online as supporting information the peer review history of all articles accepted for publication. Readers will have access to decision letters, including all Editors' comments and referee reports, for each version of the manuscript and any author responses to peer review comments. Referees can decide whether or not they wish to be named on the peer review history document.

I hope you will find the comments helpful and have no difficulty in revising your manuscript within 4 weeks.

Your revised manuscript should be submitted online using the links in Author Tasks Link Not Available. This link is to the Corresponding Author's own account, if this will cause any problems when submitting the revised version please contact us.

The image files from the previous version are retained on the system. Please ensure you replace or remove any files that have been revised. Your revised submission should include:

- A Word file of the complete text (including figure legends any Tables);
- An Abstract Figure (with legend in the Article file)
- Each figure as a separate, high quality, file;
- A full Response to Referees;
- A copy of the manuscript with the changes highlighted.
- Author profile. A short biography (no more than 100 words for one author or 150 words in total for two authors) and a portrait photograph of the two leading authors on the paper. These should be uploaded, clearly labelled, with the manuscript submission. Any standard image format for the photograph is acceptable, but the resolution should be at least 300 dpi and preferably more.

- A 'Cover Art' file for consideration as the Issue's cover image;
- Appropriate Supporting Information (Video, audio or data set https://jp.msubmit.net/cgi-bin/main.plex?form_type=display_requirements#supp).

To create your 'Response to Referees' copy all the reports, including any comments from the Reviewing Editor into a Word, or similar, file and respond to each point in colour or CAPITALS and upload this when you submit your revision.

I look forward to receiving your revised submission.

If you have any queries please reply to this email and staff will be happy to assist.

Yours sincerely,

Ian D. Forsythe
Deputy Editor-in-Chief
The Journal of Physiology
<https://jp.msubmit.net>
<http://jp.physoc.org>
The Physiological Society
Hodgkin Huxley House
30 Farringdon Lane
London, EC1R 3AW
UK
<http://www.physoc.org>
<http://journals.physoc.org>

REQUIRED ITEMS:

-Please include an Abstract Figure. The Abstract Figure is a piece of artwork designed to give readers an immediate understanding of the Review Article and should summarise the main conclusions. If possible, the image should be easily 'readable' from left to right or top to bottom. It should show the physiological relevance of the Review so readers can assess the importance and content of the article. Abstract Figures should not merely recapitulate other figures in the Review. Please try to keep the diagram as simple as possible and without superfluous information that may distract from the main conclusion of the Review. Abstract Figures must be provided by authors no later than the revised manuscript stage and should be uploaded as a separate file during online submission labelled as File Type 'Abstract Figure'. Please ensure that you include the figure legend in the main article file. All Abstract Figures will be sent to a professional illustrator for redrawing and you may be asked to approve the redrawn figure before your paper is accepted.

-Your MS must include a complete "Additional information section" with the following 4 headings and content:

Competing Interests: A statement regarding competing interests. If there are no competing interests, a statement to this effect must be included. All authors should disclose any conflict of interest in accordance with journal policy.

Author contributions: Each author should take responsibility for a particular section of the study and have contributed to writing the paper. Acquisition of funding, administrative support or the collection of data alone does not justify authorship; these contributions to the study should be listed in the Acknowledgements. Additional information such as 'X and Y have contributed equally to this work' may be added as a footnote on the title page.

It must be stated that all authors approved the final version of the manuscript and that all persons designated as authors qualify for authorship, and all those who qualify for authorship are listed.

Funding: Authors must indicate all sources of funding, including grant numbers. If authors have not received funding, this must be stated.

It is the responsibility of authors funded by RCUK to adhere to their policy regarding funding sources and underlying research material. The policy requires funding information to be included within the acknowledgement section of a paper. Guidance on how to acknowledge funding information is provided by the Research Information Network. The policy also requires all research papers, if applicable, to include a statement on how any underlying research materials, such as data, samples or models, can be accessed. However, the policy does not require that the data must be made open. If there are considered to be good or compelling reasons to protect access to the data, for example commercial confidentiality or legitimate sensitivities around data derived from potentially identifiable human participants, these should be included in the statement.

Acknowledgements: Acknowledgements should be the minimum consistent with courtesy. The wording of acknowledgements of scientific assistance or advice must have been seen and approved by the persons concerned. This section should not include details of funding.

-Author profile(s) must be uploaded via the submission form. Authors should submit a short biography (no more than 100 words for one author or 150 words in total for two authors) and a portrait photograph of the two leading authors on the paper. These should be uploaded, clearly labelled, with the manuscript submission. Any standard image format for the photograph is acceptable, but the resolution should be at least 300 dpi and preferably more. A group photograph of all authors is also acceptable, providing the biography for the whole group does not exceed 150 words.

EDITOR COMMENTS

Reviewing Editor:

The present manuscript has been reviewed by two experts in the field. Both see the potential impact of this manuscript on the field, but have also highlighted some amendments that require attention before this manuscript can be considered any further. In addition to the reviewer comments please check the following

Please check the author guidelines for symposium reviews and include an abstract figure, ideally in colour. N.b. All costs for this service, including colour reproduction in print, are covered by The Journal.

Line 17-18: is this statement of an underlying reason for circadian rhythm reflected in the data? The authors later describe how plasma lipid concentrations are elevated later in the day. Why would it make sense to have increase lipid availability overnight? An increase in NEFA concentration is an archetypal response to physical activity and stress. Is there any rationale for this overnight when sedentary? Perhaps the authors can provide a good explanation for this, or otherwise I suggest rewording to remove the notion that a physiological rhythm has any "reason" behind it

It would help if the authors are more specific with reference to which lipids they are referring to throughout the manuscript, especially as NEFA and TAG can often respond differently to various stimuli.

Line 77 - "Muscle force production" is a more accurate term than "muscle contraction". E.g. see "The function of muscle" section in DOI: 10.1080/02640410802658461

Line 95 - Should glycolysis be glycogenolysis?

Lines 107-109 - Is hepatic VLDL secretion missing from this list?

Line 112 - What is meant by a major amino acid? Most abundant? Or essential?

Line 175 - Should this read as pancreatic polypeptide?

Lines 320-332 - It might be helpful to expand on the underlying mechanisms of sleep deprivation-induced metabolic dysfunction.

Senior Editor:

This is an interesting review, but it lacks explanatory diagrams (and as you will see from published manuscripts, JP does not publish review without diagrams). Diagrams are an important way to broaden the audience for your review, they attract readers and help in the understanding of key concepts. As you revise your manuscript, please add three diagrams to your main review (introducing basic concepts, another on a key concept or two and a final one pulling together your new insights to illustrate the novelty of your ideas). This will also require some re-writing to properly integrate these diagrams into your overall narrative. Finally you will also need an abstract diagram. Please refer to past reviews to understand what is required. JP can also help with illustrations, but your essential task is to design a diagram that is informative and helps convey your key concepts.

REFeree COMMENTS

Referee #1:

Excellent paper - all comments are for clarity for a non-expert reader.]

Short title, Line 1, 'Nutrient Timing and Metabolic Regulation' needs something added - on appetite regulation and health

Please add human into the title - animal studies not covered.

Abstract - line 20/21 adjust sentence with 'both absolute and relative meal timing' as not exactly sure what this means until read later in the paper.

Line 30 - add in context it is dietary guidelines for human health; again line 186

Line 53, would be useful to guide readers to highlight, murine models have highlighted much of the mechanistic work, being applied to humans.

Line 90 - Rhythms in Macronutrient Metabolism, please confirm these all human studies being cited or note murine models.

Line 197 - how quickly does the fast eradicate the circadian rhythm in gene expression (days, hours?)

Line 204, modulators of clock gene and/or protein expression in multiple tissues- at least in murine models.

Line 249 restricting energy intake to the same set times each (time restricted feeding)

Line 262, add in a comment that these approaches are often used as a means to control body weight as a weight loss strategy or weight maintenance strategy. Also advance discussion to acknowledge that data are often collected in obese subjects and behavioural response(s) may be different to lean subjects. This is important because you are citing studies from both cohorts.

Line 264 We recently conducted a randomised controlled trial in lean participants...

Line 288 define the term physical activity thermogenesis here (usually use the term PA-EE ?) Suggest to cite

<https://doi.org/10.1111/jnc.12886> Mealtime: A circadian disruptor and determinant of energy balance?

Leonie C. Ruddick-Collins, Peter J. Morgan, Alexandra M. Johnstone

Line 361 is the first use of the word chrono-nutrition !!! Please use earlier on and suggest to cite,
<https://doi.org/10.1111/jnc.15246> Chrono-nutrition: From molecular and neuronal mechanisms to human epidemiology and timed feeding patterns Alan Flanagan, David A. Bechtold, Gerda K. Pot, Jonathan D. Johnston

Some of the references are not fully cited - presume editorial team will pick this up.

Referee #2:

This symposium review summarizes the existing literature on substrate metabolism and meal timing with circadian timing. The review also highlights the potential for nocturnal dietary interventions in improving metabolic processes. I really enjoyed reading this manuscript. The authors do an excellent job of comprehensively describing how molecular clock allows for the coordination of behavioral and metabolic cues. However, I have a few suggestions that the authors should consider addressing.

Comments

Introduction: The introduction is missing a statement of objective. The authors should also consider briefly describing the areas discussed in the rest of the review article to set up the context. Additionally, to emphasize the translational value of misalignment of rhythms in macronutrient metabolism, energy expenditure, appetite regulation, etc., authors should consider describing their potential negative impact on health.

A table briefly summarizing the proteins and genes described in the "the mammalian circadian timing system" will be helpful.

Line 127 - 141: It will be very insightful for the readers to understand how sleep phases impact basal metabolic rate at night.

Line 145 - 150: Authors are suggested to add a more comprehensive explanation of the mechanisms that explain circadian fluctuations in thermogenic effect of food and physical activity.

Line 191: Please add examples of "relative timing"

Line 232: How is "longer-term" defined in these studies on regular breakfast consumption vs breakfast skipping and metabolic adaptations?

Line 277 - 293: Are these findings comparable with the alternate-day fasting regimen where participants are not fasted completely on their "fast days"? Also, how is this comparable to time-restricted feeding?

Line 293: Please include the reference after "Overall, this work.....".

Line 310-318: I think it is also important to note that while there is a potential for nocturnal eating interventions to influence glycaemic control, an extensive literature suggests that nocturnal eating (misaligned with circadian rhythms) relates to increased risk of metabolic disorders.

I would also suggest adding a brief section on the future directions and their importance considering metabolic disorders.

END OF COMMENTS

Confidential Review

19-Sep-2021

Thank you for the careful and thorough review of this manuscript. We are very grateful for the feedback and can see that the revised version is now much more informative. Our point-by-point responses to each comment are provided below in blue italics and changes to the manuscript are indicated in red and underlined.

Editorial Comment:

Please include an Abstract Figure. The Abstract Figure is a piece of artwork designed to give readers an immediate understanding of the Review Article and should summarise the main conclusions. If possible, the image should be easily 'readable' from left to right or top to bottom. It should show the physiological relevance of the Review so readers can assess the importance and content of the article. Abstract Figures should not merely recapitulate other figures in the Review. Please try to keep the diagram as simple as possible and without superfluous information that may distract from the main conclusion of the Review. Abstract Figures must be provided by authors no later than the revised manuscript stage and should be uploaded as a separate file during online submission labelled as File Type 'Abstract Figure'. Please ensure that you include the figure legend in the main article file. All Abstract Figures will be sent to a professional illustrator for re-drawing and you may be asked to approve the redrawn figure before your paper is accepted.

Author Response: Please accept our apologies – we had prepared the required Abstract Figure but had omitted it when uploading our files. This is now included with the revised submission.

Editorial Comment:

-Your MS must include a complete "Additional information section" with the following 4 headings and content:

-Competing Interests: A statement regarding competing interests. If there are no competing interests, a statement to this effect must be included. All authors should disclose any conflict of interest in accordance with journal policy.

-Author contributions: Each author should take responsibility for a particular section of the study and have contributed to writing the paper. Acquisition of funding, administrative support or the collection of data alone does not justify authorship; these contributions to the study should be listed in the Acknowledgements. Additional information such as 'X and Y have contributed equally to this work' may be added as a footnote on the title page.

-It must be stated that all authors approved the final version of the manuscript and that all persons designated as authors qualify for authorship, and all those who qualify for authorship are listed.

-Funding: Authors must indicate all sources of funding, including grant numbers. If authors have not received funding, this must be stated.

Author Response: The required sections have been added to the title page.

Acknowledgements: Acknowledgements should be the minimum consistent with courtesy. The wording of acknowledgements of scientific assistance or advice must have been seen and approved by the persons concerned. This section should not include details of funding.

Author Response: This section was not deemed necessary since this is a review article.

-Author profile(s) must be uploaded via the submission form. Authors should submit a short biography (no more than 100 words for one author or 150 words in total for two authors) and a portrait photograph of the two leading authors on the paper. These should be uploaded, clearly labelled, with the manuscript submission. Any standard image format for the photograph is acceptable, but the resolution should be at least 300 dpi and preferably more. A group photograph of all authors is also acceptable, providing the biography for the whole group does not exceed 150 words.

Author Response: The required information is included with our resubmission.

EDITOR COMMENTS

Reviewing Editor:

The present manuscript has been reviewed by two experts in the field. Both see the potential impact of this manuscript on the field, but have also highlighted some amendments that require attention before this manuscript can be considered any further. In addition to the reviewer comments please check the following

Please check the author guidelines for symposium reviews and include an abstract figure, ideally in colour. N.b. All costs for this service, including colour reproduction in print, are covered by The Journal.

Author Response: Thank you for overseeing this review and for your comments. The Abstract Figure is now included, and we have addressed each comment below.

Line 17-18: is this statement of an underlying reason for circadian rhythm reflected in the data? The authors later describe how plasma lipid concentrations are elevated later in the day. Why would it make sense to have increase lipid availability overnight? An increase in NEFA concentration is an archetypal response to physical activity and stress. Is there any rationale for this overnight when sedentary? Perhaps the authors can provide a good explanation for this, or otherwise I suggest rewording to remove the notion that a physiological rhythm has any "reason" behind it

Author Response: Thank you for this comment. Whilst increased NEFA concentrations are indeed an archetypal response to physical activity and stress, in this scenario we believe that elevation of

lipid availability during the early evening and overnight serves to fuel energy metabolism during the longest fasting period of a given 24-hour period (i.e., sleep). This is consistent with lower respiratory quotient across the night (Zitting et al, 2018, Current Biology, <https://doi.org/10.1016/j.cub.2018.10.005>), as well as upregulated fatty acid oxidation during deep sleep (Nowak et al, 2021, Cell Reports, <https://doi.org/10.1016/j.celrep.2021.109903>). The wording in this part of the abstract is not therefore intended to infer a definitive purpose of these rhythms but rather only to state that these rhythms do facilitate alignment of nutrient availability with nutrient requirements.

Reviewing Editor: It would help if the authors are more specific with reference to which lipids they are referring to throughout the manuscript, especially as NEFA and TAG can often respond differently to various stimuli.

Author Response: Thank you for this comment, we agree that it would be more informative to be specific about the type of lipids throughout the manuscript. Where whole body lipid oxidation is referenced we have retained the broad term 'lipid' but have now specified the systemic metabolites where relevant:

Line 121-122: progressively elevated circulating non-esterified fatty acids (NEFA), triglyceride, and cholesterol later in the day and overnight

Line 127-128: circadian regulation of intestinal triglyceride absorption,

Reviewing Editor: Line 77 - "Muscle force production" is a more accurate term than "muscle contraction". E.g., see "The function of muscle" section in DOI: 10.1080/02640410802658461

Author Response: Thank you, we agree that muscle force production is a more appropriate term to use and have altered the wording accordingly.

Line 94-95: Notably, these peaks in transcript accumulation clustered at 1600 h (for genes implicated in muscle force production and mitochondrial activity)

Reviewing Editor: Line 95 - Should glycolysis be glycogenolysis?

Author Response: Thank you for spotting this error, this has now been corrected in the manuscript.

Line 114: The former is subject to endocrine regulation and driven by hepatic glycogenolysis and gluconeogenesis

Reviewing Editor: Lines 107-109 - Is hepatic VLDL secretion missing from this list?

Author Response: This is correct, VLDL appears to display diurnal rhythmicity and therefore likely contributes towards the daily variations in circulating lipids. As such this has been added to the list along with references (Marino et al, 1987; Sprenger et al, 2021).

Line 127-131: circadian regulation of intestinal triglyceride absorption, acylcarnitines, mitochondrial oxidative capacity, VLDL secretion, and insulin secretion all contributing to daily variance in blood lipid profiles (Marrino et al., 1987; Lee et al., 1992; Pan & Hussain, 2007; Ang et al., 2012; Pan et al., 2013; Yoshino et al., 2014; van Moorsel et al., 2016; Sprenger et al., 2021).

Reviewing Editor: Line 112 - What is meant by a major amino acid? Most abundant? Or essential?

Author Response: All essential amino acids display 24-hour variation. Furthermore, several non-essential amino acids (alanine, tyrosine, glycine, serine, glutamate, aspartate) and at least a couple of conditionally essential amino acids display rhythmicity also (Glutamine, cysteine) (Grant *et al* 2019 *Scientific Reports* volume 9, Article number: 4428 (2019) <https://doi.org/10.1038/s41598-019-40353->

8). The wording for this section has now been amended accordingly to reflect this along with the addition of the Grant reference above.

Line 133-134: Finally, in relation to protein metabolism, the majority of amino acids (including all essential, some non-essential and some conditionally essential) display circadian rhythmicity, with peak values occurring between 1200-2000 h and with lowest values at 0400-0800 h (Feigin et al., 1967; Wurtman et al., 1967; Feigin et al., 1968; Grant et al., 2019).

Reviewing Editor: Line 175 - Should this read as pancreatic polypeptide?

Author Response: Thank you for spotting this error, this has now been corrected in the manuscript.

Line 203: pancreatic polypeptide (peak ~1500 h, nadir ~0900 h)(Johns et al., 2006; Hill et al., 2011; Galindo Munoz et al., 2015; Rynders et al., 2020).

Reviewing Editor: Lines 320-332 - It might be helpful to expand on the underlying mechanisms of sleep deprivation-induced metabolic dysfunction.

Author Response: We agree that it is useful to provide some insight into the mechanisms linking disrupted sleep and alterations in next day metabolism. We have now added text outlining some proposed mechanisms

Line 368-372: Disrupted sleep may perturb next day metabolism through a multitude of proposed mechanisms, these include, but are not limited to: alterations in brain glucose utilization and changes in hormonal secretion profile (Scheen et al., 1996), sympathetic nervous stimulation (Spiegel et al., 2004), and/or inflammation (Meier-Ewert et al., 2004; Vgontzas et al., 2004).

Senior Editor:

This is an interesting review, but it lacks explanatory diagrams (and as you will see from published manuscripts, JP does not publish review without diagrams). Diagrams are an important way to broaden the audience for your review, they attract readers and help in the understanding of key concepts. As you revise your manuscript, please add three diagrams to your main review (introducing basic concepts, another on a key concept or two and a final one pulling together your new insights to illustrate the novelty of your ideas). This will also require some re-writing to properly integrate these diagrams into your overall narrative. Finally you will also need an abstract diagram. Please refer to past reviews to understand what is required. JP can also help with illustrations, but your essential task is to design a diagram that is informative and helps convey your key concepts.

Author Response: Thank you for these constructive comments – which have improved the content and the clarity of the review. In response to your overall feedback, we have now added 3 figures to the manuscript, the first of these outlines the basic concepts of the central and peripheral clock, as well as cellular clock machinery alongside some pertinent factors that influence these processes (e.g., light/dark, meal timing). We have also created a figure to incorporate some of these core concepts (e.g., cycles of light and dark) in order to illustrate where our body of research fits in. The final figure summarises various parameters that have been reported in different studies overnight.

REFEREE COMMENTS

Referee #1:

Excellent paper - all comments are for clarity for a non-expert reader.

Author Response: Thank you for all these helpful comments.

Referee #1: Short title, Line 1, 'Nutrient Timing and Metabolic Regulation' needs something added - on appetite regulation and health

Please add human into the title - animal studies not covered.

Author Response: We agree that the title is now more informative with the inclusion “in humans”

Referee #1: Abstract - line 20/21 adjust sentence with 'both absolute and relative meal timing' as not exactly sure what this means until read later in the paper.

Author Response: Thank you for this comment. The sentence has been revised as suggested:

Line 33-34: Experimental manipulation of feeding-fasting cycles can advance understanding of the absolute and relative timing of meals on metabolism and health.

Referee #1: Line 30 - add in context it is dietary guidelines for human health; again line 186

Author Response: Extra context to clarify that this refers to dietary guidelines for human health have been added as suggested to both lines 42 and 211.

Line 43: In summary, it is important for dietary guidelines for human health to consider nutrient timing (i.e. when we eat) alongside the conventional focus on nutrient quantity and nutrient quality (i.e. how much we eat and what we eat).

Line 217-218: Therefore, in addition to the conventional focus of dietary guidelines for human health regarding nutrient quantity and nutrient quality (i.e. how much we eat and what we eat), it is also important to consider nutrient timing (i.e. when we eat).

Referee #1: Line 53, would be useful to guide readers to highlight, murine models have highlighted much of the mechanistic work, being applied to humans.

Author Response: We agree that it is insightful to the reader to highlight that much of the mechanistic work has been performed in murine models and subsequently translated to human work. As such we have added text to this effect.

Line 68-69: Translation of murine work to humans highlights molecular regulation of circadian rhythms at a cellular level involves the expression of clock genes, which can maintain approximate 24-h rhythmicity via inter-locking transcriptional-translational feedback loops with both positive and negative limbs (Mazzocchi et al., 2012; McGinnis & Young, 2016).

Referee #1: Line 90 - Rhythms in Macronutrient Metabolism, please confirm these all-human studies being cited or note murine models.

Author Response: Throughout this section we have used a combination of human and animal work. Specifically, we have cited *in vivo* human work where possible to demonstrate rhythmicity in macronutrient metabolism with murine/*in vitro* work being used to highlight the mechanisms that may

underpin the rhythms seen. To make these broad points early in the manuscript we have cited a range of relevant papers at once and feel it would open-up a much wider discussion if we were to define whether each individual paper utilised humans or other animals, which may detract from our focus. We therefore hope that our response to the previous comment will suffice in alerting the reader at the outset to the fact that various models have been used and generalised to humans where necessary.

Referee #1: Line 197 - how quickly does the fast eradicate the circadian rhythm in gene expression (days, hours?)

Author Response: The Vollmers *et al* (2009) study subjected WT mice to a 24-h fast, which was a sufficient duration for them to observe a blunting of the circadian hepatic transcripts. For increased clarity for the reader, we have updated the text on line 227.

Line 227: yet the complete absence of daily food intake for 24-h (i.e., fasting) can eradicate the circadian rhythm in hepatic gene expression that would otherwise occur with a regular meal pattern (Vollmers *et al.*, 2009).

Referee #1: Line 204, modulators of clock gene and/or protein expression in multiple tissues- at least in murine models.

Author Response: As per above responses, we appreciate that it is more informative for the reader to clarify between human and murine models. We have therefore amended the manuscript on as per this suggestion. Furthermore, we have also added recent human evidence for insulin entraining adipose tissue rhythmicity (Tuvia *et al*, 2021, Diabetes, doi: 10.2337/db20-0910).

Line 236-237 - Indeed, feeding responsive hormones such as insulin, glucagon and IGF-1 appear to be especially potent modulators of clock gene and/or protein expression in multiple tissues – at least in murine models, however emerging evidence is now beginning to demonstrate this in humans (Tahara *et al.*, 2010; Mukherji *et al.*, 2015; Sun *et al.*, 2015; Ikeda *et al.*, 2018; Crosby *et al.*, 2019; Tuvia *et al.*, 2021).

Referee #1: Line 249 restricting energy intake to the same set times each (time restricted feeding).

Author Response: Thank you, we have amended the text on line 282 as per your feedback, with this alteration of terminology with the consideration that time restricted eating refers to human studies and time restricted feeding refers to murine models.

Line 282: Whilst skipping breakfast is often considered an unhealthy dietary approach (notwithstanding the lack of empirical support for that view), regularly omitting the same meal and/or restricting energy intake to the same set times each day (i.e., time restricted eating).

Referee #1: Line 262, add in a comment that these approaches are often used as a means to control body weight as a weight loss strategy or weight maintenance strategy. Also advance discussion to acknowledge that data are often collected in obese subjects and behavioural response(s) may be different to lean subjects. This is important because you are citing studies from both cohorts.

Author Response: Comment added line 285 and 325.

Line 284-285: By contrast, numerous other contemporary approaches to intermittent fasting, often employed as a means to control body weight through weight loss or maintenance,

Line 325-326: (it remains to be seen whether similar behavioural responses occur in obese individuals).

Referee #1: Line 264 We recently conducted a randomised controlled trial in lean participants...

Author Response: Thank you, this text has been added to line 298.

Line 298: We recently conducted a randomised controlled trial in lean participants expressly to isolate the independent effects of intermittent fasting and net energy restriction (Templeman et al., 2021a).

Referee #1: Line 288 define the term physical activity thermo here (usually use the term PA-EE ?) Suggest to cite <https://doi.org/10.1111/jne.12886> Mealtime: A circadian disruptor and determinant of energy balance? Leonie C. Ruddick-Collins, Peter J. Morgan, Alexandra M. Johnstone

Author Response: this definition has been added as follows

Line 315: physical activity energy expenditure (i.e., skeletal muscle force production (Westerterp, 2013; Ruddick-Collins et al., 2020))

Referee #1: Line 361 is the first use of the word chrono-nutrition !!! Please use earlier on and suggest to cite, <https://doi.org/10.1111/jnc.15246> Chrono-nutrition: From molecular and neuronal mechanisms to human epidemiology and timed feeding patterns Alan Flanagan, David A. Bechtold, Gerda K. Pot, Jonathan D. Johnston

Author Response: We agree that chrononutrition is a key term that benefits from being mentioned earlier in the manuscript. As such we have now mentioned and defined this term earlier, along with the recommended citation.

Line 212-213 As such, the scheduling/alignment of eating occasions (i.e., chrono-nutrition (Flanagan et al., 2021)),

Referee #1: Some of the references are not fully cited - presume editorial team will pick this up.

Author Response: Thank you - hopefully all these issues are now fixed.

Referee #2:

This symposium review summarizes the existing literature on substrate metabolism and meal timing with circadian timing. The review also highlights the potential for nocturnal dietary interventions in improving metabolic processes. I really enjoyed reading this manuscript. The authors do an excellent job of comprehensively describing how molecular clock allows for the coordination of behavioral and metabolic cues. However, I have a few suggestions that the authors should consider addressing.

Author Response: Thank you for your feedback and thoughtful suggestions.

Referee #2: Introduction: The introduction is missing a statement of objective. The authors should also consider briefly describing the areas discussed in the rest of the review article to set up the context. Additionally, to emphasize the translational value of misalignment of rhythms in macronutrient metabolism, energy expenditure, appetite regulation, etc., authors should consider describing their potential negative impact on health.

Author Response: This is an excellent suggestion – thank you. The following text has been added to the end of the introduction to set-up the ensuing sections:

Lines 58-61: The objective of this review is to briefly summarise the mammalian circadian timing system and the daily rhythmicity of macronutrient metabolism, energy expenditure and appetite regulation, before considering how the alignment of daily feeding patterns with these underlying rhythms can impact human health.

Referee #2: A table briefly summarizing the proteins and genes described in the "the mammalian circadian timing system" will be helpful.

Author Response: This has been submitted alongside the resubmitted manuscript and we have suggested that the table be included after section 1.0 on the mammalian circadian timing system.

Referee #2: Line 127 - 141: It will be very insightful for the readers to understand how sleep phases impact basal metabolic rate at night.

Author Response: Thank you, we acknowledge that this would be an insightful addition to the manuscript. We have added text to this effect on lines 163-169.

Line 163-169: Interestingly resting energy expenditure also changes overnight with differing stages of sleep (as assessed by sleep encephalography). Generally energy expenditure tends to be highest during lighter/earlier phases, and lowest during the deepest/later stages of sleep (Brebbia & Altshuler, 1965; Fontvieille *et al.*, 1994) however some studies have failed to replicate any differences between stages of sleep (Webb & Hiestand, 1975; Haskell *et al.*, 1981; White *et al.*, 1985; Palca *et al.*, 1986; Jung *et al.*, 2011).

Referee #2: Line 145 - 150: Authors are suggested to add a more comprehensive explanation of the mechanisms that explain circadian fluctuations in thermogenic effect of food and physical activity.

Author Response: We have cited the Riddick-Collins et al 2021 paper here (line 174/5) and feel that provides a really comprehensive account of the fact that the 24-h variation in TEF is completely accounted for by the underlying variation in RMR. We are therefore reluctant to speculate any further about other possible mechanisms since there may be no direct effect of time of TEF beyond that explanation. If the reviewer has something specific in mind, then we would happily consider adding that.

Referee #2: Line 191: Please add examples of "relative timing"

Author Response: Examples added to line 223

Line 223: relative timing (i.e. with respect to when other relevant events occur and/or usually occur e.g., wake/sleep, exercise, other meals).

Referee #2: Line 232: How is "longer-term" defined in these studies on regular breakfast consumption vs breakfast skipping and metabolic adaptations?]

Author Response: We understand the difficulty with this phrase here and this is why we try to avoid absolutes such as "long-term" in favour of the relative "longer-term". As such, our intention was only to contrast the duration with the previous studies that were a single day. We have now further specified the duration in question on lines 265/66.

Line 265-266: recent research has also explored the longer-term effects (i.e., 6-week) of regular daily breakfast consumption versus extended morning fasting.

Referee #2: Line 277 - 293: Are these findings comparable with the alternate-day fasting regimen where participants are not fasted completely on their "fast days"? Also, how is this comparable to time-restricted feeding?

Author Response: We appreciate the reviewer's thinking here and so had started to add new text contrasting the effects of ADF here with the past literature on other types of intermittent fasting. However, we ultimately decided against including that new content for two main reasons: firstly, to contrast each outcome (i.e. EI, FM and EI) with other studies would break-up the flow of considering how the outcomes fit together within this study; second, the other forms of fasting are different and therefore not directly comparable between trials. We hope that reasoning is acceptable.

Referee #2: Line 293: Please include the reference after "Overall, this work.....".

Author Response: Thank you, this reference has now been included and can be found on line 329.

Line 329: Overall, the data reported in Templeman *et al.* (2021a)

Referee #2: Line 310-318: I think it is also important to note that while there is a potential for nocturnal eating interventions to influence glycaemic control, an extensive literature suggests that nocturnal eating (misaligned with circadian rhythms) relates to increased risk of metabolic disorders.

Author Response: Thank you for this comment - we have added wording to that effect on lines 349-353/5.

Line 353-355: Consequently, whilst nocturnal feeding presents a possible opportunity for nutritional intervention, it also is a useful paradigm through which we can further understand the relationship between misaligned eating and the increased risk of cardiometabolic disease.

Referee #2: I would also suggest adding a brief section on the future directions and their importance considering metabolic disorders.

Author Response: Another excellent suggestion. We are very happy with the new closing line of the manuscript. Thanks again.

Lines 401-404 - Further novel insight will be possible through examining the physiological responses of human participants over complete 24-h monitoring cycles, including sequential meal tests, nocturnal feeding and with assessments under free-living conditions.

Dear Professor Betts,

Re: JP-SR-2021-280756R1 "Nutrient Timing and Metabolic Regulation Symposium Review from "Novel dietary approaches to appetite regulation, health and performance (2021)"" by Harry A. Smith and James Betts

I am pleased to tell you that your Symposium Review article has been accepted for publication in The Journal of Physiology, subject to any modifications to the text that may be required by the Journal Office to conform to House rules.

NEW POLICY: In order to improve the transparency of its peer review process The Journal of Physiology publishes online as supporting information the peer review history of all articles accepted for publication. Readers will have access to decision letters, including all Editors' comments and referee reports, for each version of the manuscript and any author responses to peer review comments. Referees can decide whether or not they wish to be named on the peer review history document.

The last Word version of the paper submitted will be used by the Production Editors to prepare your proof. When this is ready you will receive an email containing a link to Wiley's Online Proofing System. The proof should be checked and corrected as quickly as possible.

All queries at proof stage should be sent to tjp@wiley.com

The accepted version of the manuscript is the version that will be published online until the copy edited and typeset version is available. Authors should note that it is too late at this point to offer corrections prior to proofing. Major corrections at proof stage, such as changes to figures, will be referred to the Reviewing Editor for approval before they can be incorporated. Only minor changes, such as to style and consistency, should be made a proof stage. Changes that need to be made after proof stage will usually require a formal correction notice.

Are you on Twitter? Once your paper is online, why not share your achievement with your followers. Please tag The Journal (@jphysiol) in any tweets and we will share your accepted paper with our 22,000+ followers!

Yours sincerely,

Ian D. Forsythe
Deputy Editor-in-Chief
The Journal of Physiology
<https://jp.msubmit.net>
<http://jp.physoc.org>
The Physiological Society
Hodgkin Huxley House
30 Farringdon Lane
London, EC1R 3AW
UK
<http://www.physoc.org>
<http://journals.physoc.org>

Comments:

Reviewing Editor:

Thank you for addressing the comments raised in review and congratulations on an insightful manuscript.

Senior Editor:

Thank you for your attention to the minor outstanding issues. Congratulations on an interesting review.

*** IMPORTANT NOTICE ABOUT OPEN ACCESS ***

Information about Open Access policies can be found here <https://physoc.onlinelibrary.wiley.com/hub/access-policies>

To assist authors whose funding agencies mandate public access to published research findings sooner than 12 months after publication The Journal of Physiology allows authors to pay an open access (OA) fee to have their papers made freely available immediately on publication.

You will receive an email from Wiley with details on how to register or log-in to Wiley Authors Services where you will be able to place an OnlineOpen order.

You can check if your funder or institution has a Wiley Open Access Account here <https://authorservices.wiley.com/author-resources/Journal-Authors/licensing-and-open-access/open-access/author-compliance-tool.html>

Your article will be made Open Access upon publication, or as soon as payment is received.

If you wish to put your paper on an OA website such as PMC or UKPMC or your institutional repository within 12 months of publication you must pay the open access fee, which covers the cost of publication.

OnlineOpen articles are deposited in PubMed Central (PMC) and PMC mirror sites. Authors of OnlineOpen articles are permitted to post the final, published PDF of their article on a website, institutional repository, or other free public server, immediately on publication.

Note to NIH-funded authors: The Journal of Physiology is published on PMC 12 months after publication, NIH-funded authors DO NOT NEED to pay to publish and DO NOT NEED to post their accepted papers on PMC.

1st Confidential Review

03-Dec-2021